# Fostering a Wildlife-Friendly Program for Sustainable Coffee Farming: The Case of Small-Holder Farmers in Indonesia



Marco Campera [1,2], Budiadi Budiadi [3], Esther Adinda [2], Nabil Ahmad [2], Michela Balestri [1], Katherine Hedger [2], Muhammad Ali Imron [4], Sophie Manson [1,2], Vincent Nijman [1] and K.A.I. Nekaris [1,2,*]

1 Nocturnal Primate Research Group, School of Social Sciences, Oxford Brookes University, Oxford OX3 0BP, UK; mcampera@brookes.ac.uk (M.C.); mbalestri@brookes.ac.uk (M.B.); sphmanson@gmail.com (S.M.); vnijman@brookes.ac.uk (V.N.)
2 Little Fireface Project, Cipaganti, West Java 40131, Indonesia; estheradinda810@gmail.com (E.A.); ahmadnabilff@gmail.com (N.A.); katey.hedger@gmail.com (K.H.)
3 Department of Silviculture, Universitas Gadjah Mada, Yogyakarta 55281, Indonesia; budiadi@ugm.ac.id
4 Department of Forest Resources Conservation, Universitas Gadjah Mada, Yogyakarta 55281, Indonesia; maimron@ugm.ac.id
* Correspondence: anekaris@brookes.ac.uk

**Abstract:** There is an urgent need for a global transition to sustainable and wildlife-friendly farming systems that provide social and economic equity and protect ecosystem services on which agriculture depends. Java is home to 60% of Indonesia's population and harbors many endemic species; thus, managing agriculture alongside human well-being and biodiversity is vital. Within a community of ~400 coffee farmers in the province of West Java, we assessed the steps to develop a wildlife-friendly program until reaching certification between February 2019 and October 2020. We adopted an adaptive management approach that included developing common objectives through a process of stakeholder consultation and co-learning. We firstly investigated via interviews the expectations and the issues encountered by 25 farmers who converted to organic production in 2016. Their main expectations were an increase in income and an increase in coffee quality, while they had issues mainly in finding high quality fertilizers, reducing pests, and increasing productivity. We used this information to establish a problem-solving plan for the transition to community-wide wildlife-friendly practices. As part of the adaptive evaluation, we assessed the quality of coffee plantations before and after the implementation of coproduced actions. The quality of coffee significantly improved after our interventions to reduce the coffee berry borer, especially in the fields that started as inorganic and converted to organic. We uncovered additional issues to meet the standards for certification, including banning hunting and trapping activities and increasing coffee quality for international export. We describe the coproduced actions (agroforestry, conservation education, local law, organic alternatives) and phases of the program and discuss the potential barriers. We provide novel evidence of adaptive management framework successfully used to implement management actions and reach shared goals.

**Keywords:** land sharing; adaptive management; certification; organic; hunting ban; agroforestry; conservation evidence; stakeholders; implementation; co-management

## 1. Introduction

There is an urgent need for a global transition to farming systems that provide social and economic equity and protect ecosystem services on which agriculture depends [1–3]. Two main strategies have been applied in recent years to meet the growing demand for agricultural land and alleviate its impact on nature: land sparing and land sharing [2–4]. Land sparing (or nature sparing) is defined as "Increasing yields on farmed land while at the same time protecting native vegetation or freeing up land for habitat restoration elsewhere" [5]. This strategy implies high yields concentrated in a relatively small area of

land thereby allowing more efforts in the protection of nearby ecosystems. Land sharing (or wildlife-friendly farming) is defined as "Producing both food and wildlife in the same parts of the landscape by maintaining or restoring the conservation value of the farmed land itself" [5]. This strategy usually presents a lower yield per area then land sparing, thus needing a larger area for the production of the same yield. The benefit of this strategy is that wildlife-friendly farming areas contain much higher biodiversity than intensive farming areas, although they usually do not sustain the same biodiversity of the natural ecosystem [6].

Land sharing has been suggested as one of the most promising approaches to reducing deforestation in the tropics while enhancing rural livelihoods [7], although several studies suggested that land sparing is often more effective in specific contexts (e.g., [8–11]). Land sharing has been particularly effective when applied to coffee plantations since, traditionally, coffee plants were cultivated under a canopy of native trees [12]. Shade coffee plantations have been repeatedly shown to host higher biodiversity than sun exposed coffee plantations [13–16]. Shade coffee plantations may also bring other benefits such as providing alternative wildlife habitats and serving as corridors between forest fragments for arboreal mammals [17] or increasing survival of migratory birds [18]. Coffee yield also does not necessarily increase with the reduction of shade cover, rather there might be a peak in productivity at intermediate shade cover [19–21]. This relationship is because shade trees provide key services such as increasing soil quality by nitrogen fixation and increasing litter biomass, protecting from direct sun, and attracting pollinators [22].

Providing habitat for pollinators is directly linked to optimizing land use since 75% of food crops globally depend on animal pollination [23]. This is an issue at the heart of agricultural security on Java that is home to 60% of Indonesia's population and harbors many endemic species. Indonesia's National Biodiversity Action Plan calls for improving the ability of communities to conduct sustainable and equitable management of biodiversity, based on local knowledge and supported by easy access to accurate information on biodiversity functions. The Indonesian government is also sustaining organic farming and providing incentives for farmers willing to convert to organic practices.

We developed a project linking wildlife-friendly coffee production in small-holder plantations with economic security, biodiversity and ecosystem services adopting an adaptive management approach [24–26]. The adaptive management approach is a flexible approach that is based on the close connection between researchers, stakeholders, and policy makers. It involves constant collaboration and consultation between the parties involved, and regular monitoring and evaluation. As groundwork for this project, we ran ecology, oral traditions, hunting practices, conservation education and agroforestry projects with support from local communities [27–29]. We promoted wildlife-friendly practices to a community of ~400 coffee farmers in West Java since February 2019, leading them to obtain certification from the Wildlife Friendly Enterprise Network$^{TM}$ (WFEN) in October 2020 (Figure 1).

The WFEN certifies products that contribute directly to in situ conservation of key species (e.g., threats reduction), have a positive impact on the local economy and transparent community involvement. The potential market of WFEN products is specific to places that are linked to wildlife conservation (e.g., shops in zoos) The WFEN certification is an eco-certification that can help to address environmental problems and can generate economic benefits for local communities via price premiums and improved market access [30]; but see [31]. In 2020, there is only one certification (i.e., Dolphin Safe/Dolphin Friendly [32]) out of the 26 accredited in Indonesia directly linked to wildlife conservation. Here, we describe and assess the steps we followed to develop a wildlife-friendly program from its conception to reaching certification. This is a new eco-certification for the Indonesian market. We also describe the steps we took to identify problems and find common solutions with key stakeholders.

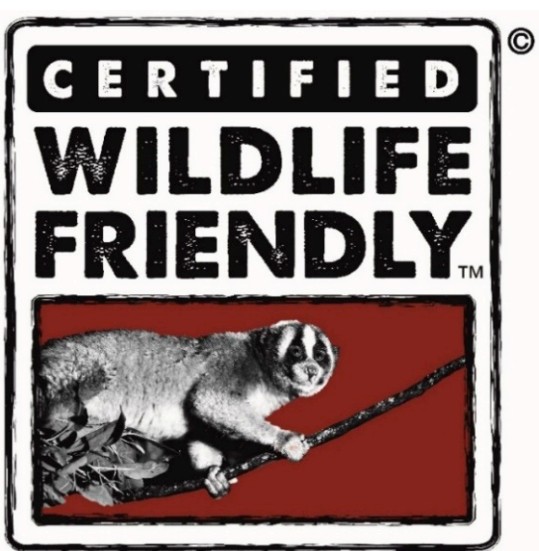

**Figure 1.** Certification obtained from the Wildlife Friendly Enterprise Network[TM] by the Little Fireface Project and coffee farmers included in the program evaluated in this study. This logo represents a Javan slow loris (*Nycticebus javanicus*), a Critically Endangered species whose conservation is central to the Little Fireface Project.

## 2. Materials and Methods

### 2.1. Study Area

We worked in the municipalities of Cipaganti and Pangauban, Cisurupan District, Garut Regency, West Java, Indonesia (Figure 2). Garut Regency is one of the biggest contributors to the agriculture sector in Indonesia [33], and a center of coffee production with 1780 tons of coffee produced per year [34]. Cisurupan district is a main producer of arabica coffee and tea (Table 1). The habitat around Cipaganti and Pangauban is a mosaic of traditional home gardens, where local farmers practice an annual perennial rotating crop system [35]. Coffee is often planted together with understory crops (e.g., cassava, chili) and shade trees. Around 400 coffee farmers are present in the area. The agroforestry system is connected to a forested area that is protected (hutan konservasi) by the Ministry of Environment.

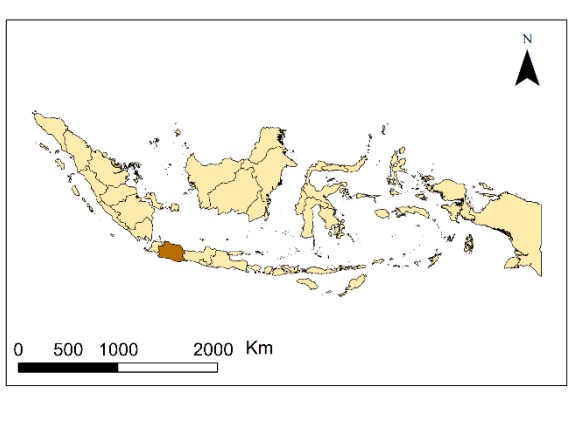

(**a**)

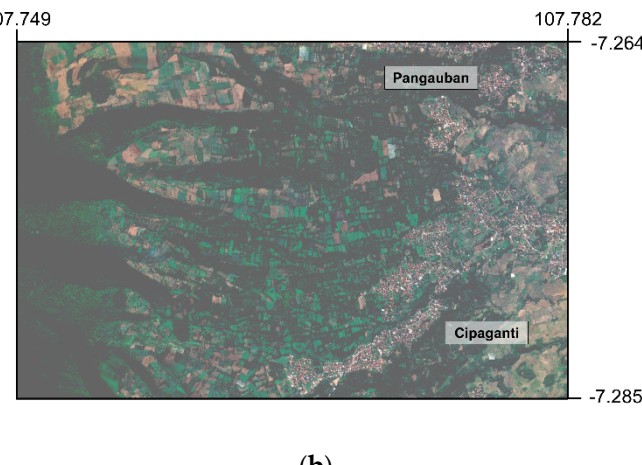

(**b**)

**Figure 2.** Location of the study area: (**a**) represents the province of West Java (Jawa Barat) in relation to the Indonesian boundary; (**b**) is a Google Earth image of the two municipalities where we delivered the program with ~400 coffee farmers. The image shows the agroforestry system connected to a protected forest. Coordinates are decimal degrees.

**Table 1.** Area under cultivation with cashcrops (in hectares, 2016) in the study district, Cisurupan, and three neighboring districts [36].

| District (Regency) | Area (km²) | People | Cultivated Land (ha) [1] | | | | | |
|---|---|---|---|---|---|---|---|---|
| | | | *Arabica* | *Robusta* | **Tea** | **Tobacco** | **Vetiver** | **Sugar Palm** |
| Cisurupan (Garut) | 8088 | 107,046 | 239 | 67 | 219 | 47 | | 8 |
| Samarang (Garut) | 5971 | 77,833 | 155 | 46 | | 210 | 1020 | |
| Bayongbong (Garut) | 4763 | 104,938 | 38 | 9 | 8 | 313 | 272 | 16 |
| Cikajang (Garut) | 12,495 | 90,173 | 365 | 59 | 286 | | | 86 |

[1] Arabica: *Coffea arabica* L.; robusta: *Coffea canephora* Pierre; tea: *Camellia sinensis* (L.) Kuntze; tobacco: *Nicotiana tabacum* L.; vetiver: *Chrysopogon zizanioides* (L.) Roberty; sugar palm: *Arenga pinnata* (Wurmb.) Merr.

### 2.2. Approach to Conservation Implementation

We adopted an adaptive management approach that included developing common objectives through a process of stakeholder consultation and co-learning (Figure 3). Stakeholders included key coffee farmers, members of the local governments of Cipaganti and Pangauban, members of the Balai Konservasi Sumber Daya Alam (BKSDA—Natural Resources Conservation Agency, branch of the Ministry of Environment), and members of the Dinas Perkebunan (Disbun—Department of Agriculture). Adaptive management is a key approach in conditions of uncertainty and rapid environmental and social change [24], thus ideal for our program. This approach involves cycles of adaptive governance where we co-identified problems with stakeholders, adaptive planning where we coproduced management actions with stakeholders, and adaptive management where we designed and implemented the coproduced actions. At all the stages of the cycles there is an adaptive evaluation, meaning that we learn by doing and we always consider feedback from the different stakeholders to rethink and replan actions.

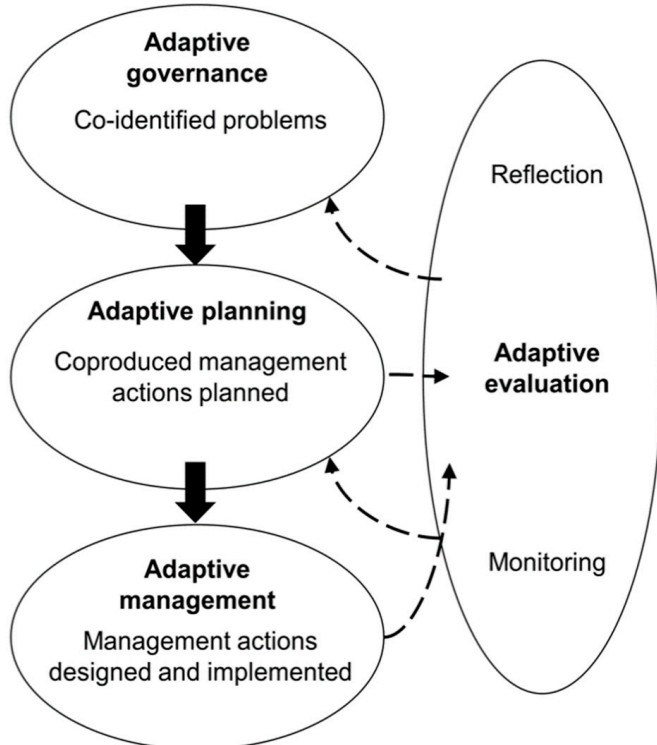

**Figure 3.** Adaptive management framework applied to establish a problem-solving plan for the transition to community-wide wildlife-friendly practices in ~400 coffee farmers in West Java, Indonesia.

During the adaptive governance stage, we co-identified problems with stakeholders. We started the adaptive governance stage in February 2019. We interviewed a group of 25 local farmers (14 men and 11 women) from Pangauban village who converted to organic farming in 2016 to begin a process to obtain certification based on Indonesian standards. They obtained the certification in October 2019, thus after our first survey. We asked 13 open-ended questions (Table 2) to obtain background information on their agricultural and ecological knowledge, the existing coffee farming and selling systems in place in the area (e.g., to identify key farmers), perception of the organic project, and expectations from converting to organic farming. We used the information collected as a first step to identify potential problems for the transition to community-wide use of wildlife-friendly practices. The interviews were done in Sundanese (local language) by a local assistant with a bachelor's degree in education and with previous experience in doing surveys. The adaptive governance stage was regularly repeated during the program though regular meetings and focus groups with key farmers and stakeholders to co-identify problems that could have emerged during the process.

**Table 2.** Open-ended questions used in the interviews with 25 coffee farmers in Pangauban village, West Java. We also took note of sex and age of participants.

| N | Question |
|---|---|
| 1. | Which crops do you have? If more than one, which one is more important? |
| 2. | Where does your main income come from? Is this income from farming? |
| 3. | How much do you earn yearly from farming coffee? |
| 4. | At what price do you sell coffee per kg and to whom do you sell it? |
| 5. | Do you still use chemical pesticides or herbicides? If so, which one? |
| 6. | What fertilizers do you use? |
| 7. | Have you ever attended a training on how to produce high-quality fertilizers? |
| 8. | Do you have a problem with your crops? If they say pest, ask to explain it in detail. |
| 9. | How do you use local natural resources and wildlife? If they say hunted, trapped or pest, ask to explain in detail. |
| 10. | Do you think that wildlife plays an important role in coffee production? |
| 11. | What do you know about Fairtrade, organic and wildlife-friendly agricultural certifications? |
| 12. | Are you willing to participate in seminars, training on Fairtrade, organic and wildlife-friendly certification? |
| 13. | What did you expect from converting to organic and what do you expect now from our project? |

During the adaptive planning stage, we coproduced management actions. We considered several elements to decide which actions to implement, including costs, availability, and efficiency. We mainly acted as consultants and scientific advisors, providing several alternatives and discussing their applicability with stakeholders. We prioritized stakeholders' opinions and values in the choices of the best alternatives [26]. This is to favor their acceptance of the actions taken and the long-term sustainability of the program [26]. The same principles were considered during the adaptive management stage when we designed and implemented the coproduced actions.

During the adaptive evaluation stage, we reflected on and monitored the co-identified problems and co-produced actions. This process involved a cycle of meetings and focus groups with the different stakeholders, as well as independent monitoring of the actions to check if they were implemented by farmers. As part of the adaptive evaluation, we assess the quality of coffee plantations before and after the implementation of co-produced actions. We monitored 28 organic fields (belonging to the 25 farmers included in the interviews) and 28 inorganic fields in terms of infestation of coffee berry borers. The total area surveyed was 3.4 ha in organic fields and 3.7 ha in inorganic fields. The coffee berry borer (*Hypothenemus hampei*) is the main cause of yield loss and reduction of fruit quality for coffee plants throughout the world [37]. We collected data in two ripe fruiting seasons May 2019 and May 2020 (before and after the implementation of coproduced actions; Table 3) on the proportion of branches damaged by coffee berry borers. We sampled 10 random coffee plants in each field and gave a score from 0 (no branches damaged) to 10 (all the branches highly damaged), following [38]. We calculated a mean score for each field.

**Table 3.** Actions taken by the ~400 coffee farmers in Cipaganti and Pangauban, West Java, at different stages of the program. Some actions were already present before the program started, other actions were implemented after different phases of stakeholder consultation and co-learning.

| Action Group | Pre-Consultation | Phase 1 | Phase 2 |
|---|---|---|---|
| Agroforestry | 1. In 2014, the Little Fireface Project launched a pilot agroforestry project with a plant nursery that provided ~2000 trees to the farmers. 2. Between 2016 and 2019, the Little Fireface Project installed waterline bridges to improve connectivity for mammals and provide irrigation for most fields [29]. | 1. We identified the need to have more shade trees since several plantations were sun-exposed and that limited the connectivity for animals. 2. We identified several native species to use as shade trees in coffee plantations so that soil quality would improve, and to attract pollinators. | 1. The farmers obtained ~30000 trees from the government. These trees species were from the list of selected trees identified in Phase 1 (Appendix A). |
| Education and wildlife conservation | 1. Between 2014 and 2019, the Little Fireface Project delivered several conservation education programs meant to increase knowledge about wildlife, with a particular focus on the Critically Endangered Javan slow loris [27]. 2. Thanks to the work of the Little Fireface Project, the hunting and trapping activities were already reduced, and hunting and trapping in the area was very limited. | 1. We delivered a specific program with schools in the village and neighboring areas to promote the importance of wildlife in coffee plantations. 2. During the meetings with the farmers, we consistently emphasized the importance of wildlife in coffee plantations and the benefits of having shade trees and pollinators. 3. The farmers suggested that a local law (i.e., a community agreement with fines for offenders) would be a good solution to stop the limited hunting and trapping activities in the area. | 1. We organized a regional workshop with farmers from West and Central Java, representatives from the Government, NGOs, and academics working on sustainable farming production (>160 participants). 2. The local law banning hunting activities was approved in January 2020 and the community wanted also to address the issue of waste management in the law. Therefore, a littering ban is now in place and a waste management plan is in development. We placed a large sign to the entrance of the village and 15 medium signs at the entrance of each unpaved road used to access the agroforestry area. |
| Organic alternatives | 1. A group of 25 farmers started a program for organic certification in 2016 and obtained some benefits from the government. 2. Organic farming is a general practice in the area, with a general good knowledge about organic fertilizers | 1. We organized training for local farmers to increase their knowledge and use of high-quality organic fertilizers. 2. We provided equipment for the farmers who joined the program. We guaranteed that the farmers did not have additional costs after their shift to wildlife-friendly practices. | 1. The farmers asked the government for additional training regarding organic pesticides and fertilizers. 2. We tested, together with farmers, several organic pest control methods and applied the best method to all the fields before each fruiting season. This resulted in a significant decrease in coffee berry borer infestation (Figure 5). |

### 2.3. Statistical Analysis

To evaluate the first step of the adaptive governance stage (i.e., interviews), we have additionally tested for mediation effects between variables via structural equation modelling (SEM) via IBM Amos 26 software. In this analysis, we used the variables that emerged from the questionnaire sex (women = 0, men = 1), age (continuous variable), perceptions on what benefits wildlife brings (no = 0, yes = 1) and "income increase is the only benefit wanted" (i.e., farmers only indicated an increase in price of their coffee as expected benefit) (no = 0, yes = 1), as both dependent and independent variables (exogenous variables), mediating the variables "use pesticides" and "hunt or ask to hunt pests" (endogenous variables). We used maximum likelihood estimation and bias corrected 95% confidence intervals to calculate model parameters. We assessed the goodness-of-fit of

our model by chi-square ($\chi^2$) test, root mean square error of approximation (RMSEA) and comparative fit index (CFI; [39,40]).

We tested the efficiency of the co-produced actions meant to improve the quality of coffee berries by evaluating whether the coffee berry borer infestation score was lower after the interventions. We ran a generalized linear mixed model with coffee berry borer infestation score as dependent variable, coffee field as random factor, stage (before and after intervention) as repeated measure, and farming type (organic vs. inorganic) as fixed effect. We transformed the data to quasibinomial distribution and used the function "glmmPQL" in the package "MASS" via R v 4.0.3.

## 3. Results

### 3.1. Co-Identified Problems

From the interviews with the farmers, we initially identified potential problems and the potential expectations of farmers in shifting to wildlife-friendly farming. The 25 farmers from Pangauban village started using organic practices in 2016 and three years after (at the time of the interviews) they had not yet obtained the certification. They identified the following problems encountered during the conversion to organic farming: decrease in productivity (36%), high cost and long time to produce organic fertilizers (36%), difficulties in removing the main pest of coffee plantations, the coffee fruit borer (28%), and high costs and labor time to remove weeds (8%). Additionally, they had the following expectation in converting to organic: higher income (80%), help with organic fertilizers (32%), increase in coffee quality (24%), help with irrigation systems (12%), help from the government (4%), training (4%), and a waste management plan (4%). Some of the farmers (16%) declared to have used chemical pesticides and fertilizers since they did not see a short-term benefit from converting to organic, rather they found a decrease in productivity and in coffee quality. Some of the farmers (12%) declared to have hunted or asked to hunt perceived pests (e.g., Javan palm civet *Paradoxurus musangus javanicus* [41]) seen feeding among their crops. From the SEM (Model fit: $\chi^2$ = 8.166, *p* = 0.318, RMSEA = 0.083, CFI = 0.902), we found that they declared to have used chemicals and to have hunted or asked to hunt more if they were men and if they did not think that wildlife gives benefits (Figure 4). Co-identification of problems continued for the whole duration of the program together with the other phases based on the adaptive management framework. This involved regular bi-weekly meetings between researchers and key farmers, and regular monthly meetings with the other stakeholders.

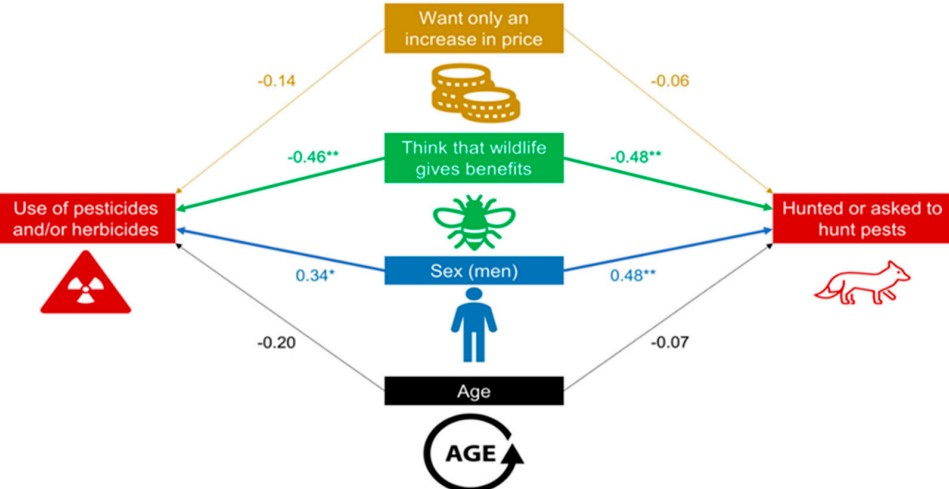

**Figure 4.** Representation of the structural equation model to understand the determinants of use of pesticides and/or herbicides and hunting of and requests to hunt pests (endogenous variables) in 25 coffee farmers is Pangauban, West Java. Thick arrows starting from the exogenous variables indicate significant relationships. * *p* < 0.05; ** *p* < 0.01.

### 3.2. Management Actions

We coproduced management actions at different stages of the project to obtain the certification (Table 3; Figure 5). Phase 1 refers to the time (February-June 2019) between the beginning of the program and the general agreement between all the farmers to join the program. Phase 2 refers to the time (July 2019–October 2020) until the certification was obtained.

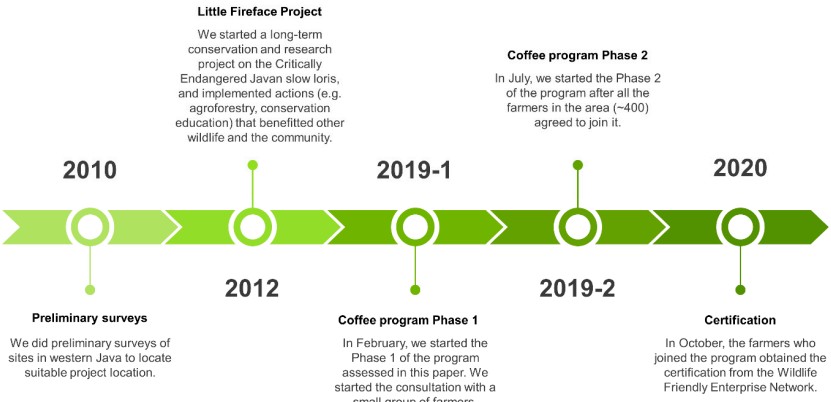

**Figure 5.** Timeline showing the progress of the program assessed in this paper and the background work done since 2010 (pre-consultation stage). More details are in Table 3.

The coffee berry borer infestation was reduced after the implementation of the co-produced actions (stage effect: $\beta = -1.51 \pm$ SE 0.22, t-value = $-6.96$, $p < 0.001$) (Figure 6). Organic plantations had significantly higher infestation than inorganic plantations (farming type effect: $\beta = 0.76 \pm$ SE 0.24, t-value = 3.17, $p = 0.003$), and this was dependent to the fact that the reduction in infestation was more evident in the plantations that started as inorganic and turned into organic (stage*farming type interaction effect: $\beta = 0.77 \pm$ SE 0.29, t-value = 2.69, $p = 0.009$). The initial value of the infestation, in fact, was similar between organic and inorganic plantations (Figure 6).

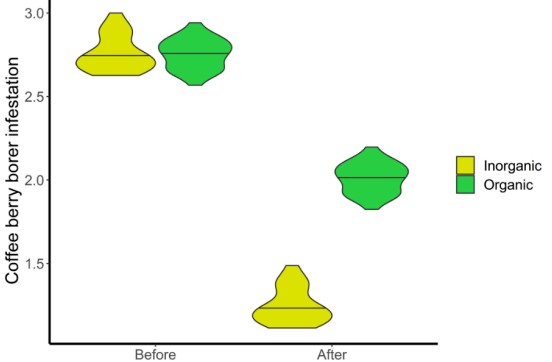

**Figure 6.** Violin plot representing the decrease in coffee berry borer infestation in 28 organic and 28 inorganic coffee fields in Cipaganti and Pangauban, West Java. We did the assessment before (May 2019) and after (May 2020) we implemented the coproduced actions meant to reduce the coffee berry borer. The inorganic plantations turned into organic during the process, starting in July 2019. Filled areas indicate density of data, lines indicate median values.

## 4. Discussion

### 4.1. Co-Identified Problems

To achieve certification from the WFEN, coffee farmers must satisfy several conditions that include, but are not limited to, establishing a ban on hunting and trapping activities in the immediate area, processing that meets international export standards, transparent community involvement, and using organic farming methods. The establishment of these requirements relies heavily on the support of different stakeholders, and the adaptive man-

agement framework was an effective and flexible approach to use in our program [24]. In fact, we needed a flexible approach with several cycles of problem definition, planning, and implementation as our program involves multiple stakeholders and knowledge exchange. Our program adds to the list of the few successful programs (8% out of 187 programs based on [25]) that implemented management actions after using an adaptive management framework. The regular meetings with coffee farmers and other stakeholders were essential to co-identify problems and co-produce solutions. For example, the farmers suggested that the implementation of a local law banning hunting and trapping and littering would have been feasible and would have provided long-term benefits to the community. Another important action suggested by the farmers was the request to the government for additional training, equipment, and shade trees (Table 3). Local farmers surely know regulations and development opportunities more extensively than foreigners, thus providing an invaluable help in community development programs. In addition, local farmers had their own network of consultants, experts, and government authorities and they allowed us to join that network. This was, in our opinion, one of the main advantages of adopting the adaptive management framework. Farmers felt empowered by the collaboration with us and further expanded their network, with the consequent expansion of our network. We have to note that a key factor driving to the success of our program was the long-term conservation effort and community involvement in the study area by the Little Fireface Project [27]. This favored a positive involvement of the local farmers that showed pro-environmental behaviors since the beginning of the program. We also need to consider that the incentives to support wildlife-friendly practices may not be as obvious to those who are not aware of the direct and indirect benefits wildlife-friendly certifications bring to local communities. Interviewed farmers used significantly less chemicals and hunted/trapped or asked to hunt/trap less if they thought that wildlife gives benefits. We thus needed to focus our attention on the value of wildlife-friendly farming and the consequent increase in crop quality (expected benefit from turning into organic) that the farmers will obtain from it.

Wildlife-friendly farming programs often provide monetary incentives for farmers, referred to as incentive programs or payments for ecosystem services [42]. Although the program we developed provided useful tools to farmers such as sickles, modes of producing fertilizer and pest control materials, we did not provide direct, monetary compensation for maintenance costs incurred during the conversion to wildlife-friendly farming. Previous research into the alignment between farmer values and wildlife-friendly initiatives has shown that it is the values of the initiative itself that promotes farmer participation, not the subsequent monetary compensation, if supplied [42]. Therefore, our priority was ensuring that the values of our program align with those of farmers, so as not to alienate their local culture and traditions. This can be done through identifying which conservation methods they support, recognizing how they refer to the program in terms of the language used and ensuring they are included when establishing the "rules" and "values" of the program [42].

### 4.2. Management Actions

#### 4.2.1. Agroforestry

There is much debate as to the effectiveness of agroforestry in the preservation of biodiversity, comparing the viability of land-sparing versus land-sharing. The majority of these comparisons place land-sparing above land-sharing in terms of preserving biodiversity [8–11], yet often these studies do not take into account existing conditions and limitations. In West Java, smallholder agricultural plots dominate the landscape outside of conservation and protected areas. These agroforest environments bordering protected areas render land-sparing redundant as much of the landscape has already been deforested and would not provide good habitat if included in a land-sparing initiative. Additionally, studies have found that in small-holder dominated landscapes, only those farmers who have significant social, personal and financial capital will benefit from land-sparing, therefore the distribution of wealth will be skewed [43]. Instead, this landscape poses an interesting opportunity to gauge the viability of land-sharing within an agroforest environment.

In some case studies where land-sharing proved to aid conservation efforts, it was found to only be beneficial for animals and plants that provided direct benefits to local farmers, so overall biodiversity was not preserved, whilst land-sparing has been found to benefit species that are in most need of conservation regardless of the ecosystem services they provide [5,44]. However, the argument of land-sharing vs. land-sparing is redundant as the success of each depends entirely on their setting, and furthermore, that setting's history, both recent and long-term [7,43]. Although land-sparing has garnered more support by researchers, particularly in regard to bird biodiversity [10,11,45], conservationists in Java have little choice but to consider land-sharing due to the lack of natural forest remaining after centuries of deforestation [46]. Agroforestry provides wildlife with viable habitat within farming landscapes and aims to combine wildlife-friendly farming practices with conservation efforts, without the need for extra land [7,43]. Therefore, due to the limitations regarding natural land in West Java, agroforestry is the most viable option after examining the trade-offs between economic prosperity and biodiversity preservation [43].

Agroforest environments protect biodiversity in areas of intensive farming by providing connections essential to the movement and subsistence of local wildlife [7,47]. Planting 'living fences' alongside farms allows wildlife to persist and even thrive [35,48]. By providing corridors, home-ranges are preserved, and the provision of pollination, seed dispersal and pest control services by vertebrate species is encouraged [29]. These 'living fences' must be created with local wildlife and local agriculture in mind as the integration of certain tree species can be especially beneficial to the surrounding ecosystem (Appendix A). For example, suren trees (*Toona sureni* (Blume) Merr.) help to stabilize soil, protect areas of human habitation against landslides and promote natural, ecological relationships that encourage the provision of ecosystem services such as maintaining the natural enemies of pests [49,50]. They can also be farmed for their wood and leaves, providing extra income for farmers [49,50]. As well as encouraging farmers to plant other crops to increase income, plant diversity and canopy cover within their plantations, one of the coproduced actions was to obtain tree saplings of selected trees from the government to improve local agroforestry systems. Whilst agroforestry protects ecosystem services provided by local wildlife, increased canopy cover increases the quality of coffee grown underneath, protects crops during extended periods of dry weather and hosts higher biodiversity [13–16]. Higher biodiversity allows for more sustainable pest management, as higher species richness provides more even pest predation than if only one dominant species were to provide this service alone [51].

### 4.2.2. Education and Wildlife Conservation

A key part of our program was to show that wildlife can benefit farmers, a key factor to consider when promoting wildlife-friendly farming practices as highlighted from the results of the interviews. Vertebrate species provide key pollination, seed dispersal and pest control services in the tropics and help to produce economically important crops, such as arabica coffee [52,53]. Birds predate on pests and pollinate coffee plants [54], the critically endangered Javan slow loris eats pests and pollinates shade trees and Javan palm civets act as seed dispersers. In conservation terms, it is also important that hunting bans are issued in agroforest environments as deforested areas open up opportunities for hunting due to increased accessibility [44]. The ecosystem services vertebrates provide will enable coffee farmers to charge premium prices for their coffee due to its certified status and the quality of the coffee, which increases when grown using wildlife-friendly methods, and this increased price will offset the potential initial reduction in productivity that comes with agroforestry [43,55]. Therefore, preserving native plant and wildlife species benefits both local conservation efforts and the social and economic wellbeing of local communities. While education and discussions will make up a large part of our approach to prevent hunting activity in the area, it is important to establish other activities that hunters can partake in. In order for this to be a sustainable, long-term program, significant decisions

such as these should be made by the stakeholders involved, therefore farmers should come to this conclusion themselves.

In West Java, trapping is common practice, especially for songbirds, small carnivores and slow lorises for the pet trade [56–58]. Although there has been a general decrease in bird hunting and trapping activities in recent years, this is only due to the decreasing availability of commercially sold birds in response to the marked overexploitation that has occurred over the last decade due to the pet trade [59]. Alongside hunting and trapping for commercial purposes (e.g., civets for coffee factory, kopi luwak), hunting is also a popular hobby amongst local people [60]. In the pursuit of wildlife-friendly certification, there should be no hunting or trapping in the surrounding area. After months of discussions with farmers, hunters and members of the local government, the community introduced a hunting and littering ban into local law. The hunting and trapping ban is redundant to national regulations as these already make it illegal to hunt any species of animal, including those that provide economically important ecosystem services such as birds, slow lorises and civets. The local law, however, is expected to put more pressure on the hunters (including those that merely hunt as a hobby) and trappers and provide increased enforcement as the community can feel empowered to stop hunters and trappers and the signs that are in the area can deter hunters from accessing traditional hunting sites. In order for this law to be successful, farmers and local residents must be aware of the benefits of wildlife to local agriculture and be able to organically come to the conclusion that hunting and trapping directly jeopardizes this. Therefore, education and outreach were important parts of this process. By holding meetings with farmers, we were able to communicate the importance of biodiversity and following this, farmers conceived and supported the idea of a hunting ban themselves. Regular meetings and co-management are key parts of interventions aiming at sustainability in agroforestry areas (e.g., [61]). Going forward, the importance of biodiversity will have to be reinforced through regular meetings with farmers. As of yet, hunting and/or trapping activity has not been monitored closely; in order for this to be sustainable, laws must be properly regulated, and penalties must be enforced.

Obtaining certification is a lengthy process, for both wildlife-friendly and organic certifications, with significant milestones often going unseen. Farmers will become less-invested when the process becomes too time-/resource-consuming. It is important that requirements from WFEN certification are pushed through quickly to show farmers that progress is being made and to retain momentum. With the inauguration of the hunting and littering ban, waste management has become a major priority of the coffee program. Barriers to proper waste management exist in many parts of the world and obstruct access to social, economic and environmental benefits [62,63]. In partnership with local government and local recycling centers, we have designed methods to promote proper waste management. These methods are twofold: education programs for the local community on both the practical aspects and environmental benefits of recycling and provision of the appropriate infrastructure to deal with the waste produced.

Whilst the education program for the local community regarding the importance of recycling in the preservation of natural environments is crucial, prioritizing this over establishing infrastructure is not recommended as research shows that investing in the practical aspects of the process are more important when implementing pro-environmental practices [63,64]. If the infrastructure exists, residents will be more inclined to contribute to the pro-environmental practices regardless of their original attitude. However, the importance of education should not be dismissed entirely. Research shows a clear correlation between the knowledge of the environmental benefits of recycling and consistent recycling habits [62,65]. Therefore, if we are to promote long-term recycling habits, it is essential that waste management is included in local school curriculums.

### 4.2.3. Organic Alternatives

Chemical fertilizers and pesticides are generally favored due to their low cost and high initial productivity. Not only is this productivity often short-lived, but the consistent

use of chemicals such as these threaten the health of farmers and ultimately, the health of the surrounding ecosystem [66,67]. The prolonged use of chemical fertilizers changes the pH of soil, causing it to become too acidic, and can also impair soil fertility [68,69]. The reversal of soil acidification and restoration of soil fertility is possible through the use of natural fertilizers [68] but requires not only the commitment of the farmer/s in question, but every farmer in the area due to the possibility of run-off. Natural fertilizers, such as chicken manure, are already used due to their wide availability and low cost. However, for farmers to convert to solely natural fertilizers, there must be a compromise between cost-effectiveness and organic nature. The interviewed farmers highlighted that organic fertilizers are often low-quality and their preparation is time-consuming and expected help with organic fertilizers. Rabbit manure is a viable option for an alternative natural fertilizer as it has been found to be more cost-effective than inorganic fertilizers [70] and twice as rich in nutrients as chicken manure [71]. EM4, an effective microorganism used to boost plant growth, is applied to manure to increase the rate of decomposition and to induce higher nitrogen uptake by the crop [72]. Several studies have found EM4 to be beneficial to plant growth when used in combination with organic and inorganic fertilizers, yet there have been no studies analyzing the efficiency of EM4 in the presence of organic fertilizer alone. Although productivity may initially be lower due to the exclusion of inorganic fertilizers, wildlife-friendly farming has been shown to increase crop yield in the long-term [73].

One of the biggest problems in promoting organic and wildlife-friendly practices is encouraging the use of natural pesticides in the face of increasing demand for high crop productivity [51]. Experiments have been carried out to establish which of the available natural pest controls are the most cost-effective [74]. Hypotan and Glumon are two examples of organic pheromone-based pest control available in Indonesia. Hypotan draws the coffee fruit borer into a pool of water and Glumon catches the borer on its surface, and it is suggested by the farmers to be more efficient. Through the coffee program, we provided these organic pest control solutions to farmers prior to harvest in place of chemical pesticides which are commonly used in Indonesian small-holder plots. As well as moving farmers closer to certification, natural pest control methods require lower input costs and can be extended to natural pest control services from wildlife [75]. The fact that our coproduced actions brought to a significant reduction of coffee berry borer infestation indicate the importance of the adaptive management framework. The reduction was significantly higher in the plantations that started as inorganic and turned into organic. The pest control actions were thus particularly effective to persuade farmers in maintaining organic practices as they can see tangible results. The farmers that started as inorganic were even more enthusiastic as they perceived a clear increase in the quality of their coffee without a loss in productivity. It is also important to understand what is locally available and what can be more accepted locally so that farmers will use what is provided.

## 5. Conclusions

We successfully applied an adaptive management framework to foster a wildlife-friendly program for coffee farmers in Java, Indonesia. By using this framework, we co-identified specific problems and co-produced successful actions that were fully accepted by the community and brought to significant improvements for the community (e.g., decrease in coffee berry borer infestation). Surely, the success of the program was also derived by the long-term conservation effort and community involvement that was done in the area since 2012 [27]. Our program represents one of the few studies that declared to have used this framework and implemented management actions to reach a shared goal; in our case, certification to promote a long-term sustainability [24,25]. This success is probably linked to the crop type (i.e., coffee) and region of study (i.e., Java) as coffee represents a crop that can benefit from tree shade and the connectivity given by an agroforestry system, and Java is a highly populated highland where the original forest is highly fragmented. Thus, promoting wildlife-friendly agroforestry systems is the best solution to ensure the long-term sustainability of both biodiversity and the livelihoods of local farmers. Co-

produced actions to reduce the pest control and use high quality organic fertilizers were particularly effective to persuade farmers in joining our program and need to be continued in a regular way. The fact that we co-produced actions with farmers, and that we involved other stakeholders from the Government and Universities was key to ensure participation to our program. The farmers perceived our interest in improving their productivity by providing accessible solutions and consultation with local experts. As a final step to ensure the long-term sustainability, we need to ensure that the farmers keep their involvement and maintain their understanding of the value of the certification. This would require constant actions at least until there is a system in which farmers will obtain a premium price for exporting their coffee (80% of coffee farmers expected higher incomes from turning into organic, as shown by the results of our interviews), a benefit often missing after obtaining an eco-certification [31].

**Author Contributions:** Conceptualization, M.C. and K.A.I.N.; methodology, M.C.; software, M.C.; validation, M.C. and K.A.I.N.; formal analysis, M.C.; investigation, M.C., E.A., N.A., M.B., K.H. and S.M.; resources, B.B., M.A.I., V.N. and K.A.I.N.; data curation, M.C., M.B., K.H., S.M. and K.A.I.N.; writing—original draft preparation, M.C. and S.M.; writing—review and editing, B.B., E.A., N.A., M.B., K.H., M.A.I., V.N. and K.A.I.N.; visualization, M.C.; supervision, B.B., M.A.I. and K.A.I.N.; project administration, K.A.I.N.; funding acquisition, M.B. and K.A.I.N. All authors have read and agreed to the published version of the manuscript.

**Funding:** This research was funded by Augsburg Zoo, Brevard Zoo, Cleveland Zoo and Zoo Society, Columbus Zoo and Aquarium, Disney Worldwide Conservation Fund, Global Challenges Fund Initiative—Oxford Brookes University, International Primate Protection League, Lee Richardson Zoo, the Leverhulme Trust (RPG084), Mohamed bin al Zayed Species Conservation Fund (152511813, 182519928), Margot Marsh Biodiversity Fund, Moody Gardens Zoo, National Geographic (GEFNE101-13), NaturZoo Rhein, People's Trust for Endangered Species, Primate Action Fund, San Francisco Zoo, and Shaldon Wildlife Trust.

**Institutional Review Board Statement:** The study was conducted according to the guidelines of the Declaration of Helsinki, and approved by the Oxford Brookes University Ethics Committee (number 181256, 1 February 2019).

**Informed Consent Statement:** Informed consent was obtained from all subjects involved in the study.

**Data Availability Statement:** The data presented in this study are available on request from the corresponding author. The data are not publicly available due to privacy reasons.

**Acknowledgments:** We thank Indonesian RISTEK for authorising this study. We thank our field team D. Ahmad, A. Ardiansyah, H. Birot, T. Busina, R. Cibabuddthea, R. Hidayat, G. Himawan, Y. Nazmi, A. Nunur, D. Rustandi.

**Conflicts of Interest:** The authors declare no conflict of interest. The funders had no role in the design of the study; in the collection, analyses, or interpretation of data; in the writing of the manuscript, or in the decision to publish the results.

## Appendix A

**Table A1.** List of shade trees that we suggested to the coffee farmers since native to the study area and with benefits for their crops and/or for the community.

| Scientific Name | Vernacular Name | Benefits |
|---|---|---|
| *Aleurites moluccanus* (L.) Willd. | Kayu kemiri | Edible parts, medicine, habitat restoration |
| *Altingia excelsa* Noronha | Rasamala | Edible parts, increase soil quality, medicine |
| *Averrhoa bilimbi* L. | Belimbing | Edible parts, medicine, timber |
| *Cinnamomum burmannii* (Nees & Th. Nees) Blume | Kayu manis | Construction, edible parts, essential oils, medicine, timber |
| *Elaeocarpus ganitrus* Roxb. | Ganitri | Medicine, traditional use in religion |
| *Eusideroxylon zwageri* Teijsm. & Binn. | Kayu besi | Increase soil quality, timber |
| *Ficus padana* Burm.f. | Kayu hamerang | Edible parts, increase soil quality, medicine |
| *Hibiscus tiliaceus* (L.) Fryxell | Kayu waru | Construction, edible parts, medicine, timber |
| *Manglietia glauca* Blume | Manglid, baros | Construction, timber |
| *Melaleuca cajuputi* Powell | Kayu putih cirebon | Essential oils, medicine, timber |
| *Syzygium polyanthum* (Wight) Walp. | Kayu salam | Edible parts, medicine, timber |
| *Toona sureni* (Blume) Merr. | Suren | Construction, increase soil quality, medicine, timber |

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
