# Peer review of "Fostering a Wildlife-Friendly Program for Sustainable Coffee Farming: The Case of Small-Holder Farmers in Indonesia"

_land, doi:10.3390/land10020121_

Round 1
Reviewer 1 Report
I enjoyed this paper on your project that integrates ecosystem services with agricultural production, demonstrating how wildlife-friendly coffee plantations can effectively protect biodiversity. I have included comments in the attached pdf with corrections and recommendations. My main concern is that the emphasis in the method and result sections on the data analysis from the interviews was barely touched upon in the discussion, and not at all in the conclusions. How does the results of the interview show the benefits of having wildlife on the farm for coffee production? I don't see in this paper these specific results nor do I see where this study actually demonstrated that wildlife benefited from this program. Furthermore, the discussion needs to cover the evolution of the adaptive management process, lessons learned and recommendations to make the adaptive management process successful. You had mentioned that a majority of adaptive management programs had failed, while yours was successful. What did you do differently that made for your success? Did this case study of 25 farmers who went organic and did not hunt vertebrate pests show us that this method improved crop quality? What were the monetary costs to the farmers? How much did they lose to gain a biodiverse ecosystem? I didn’t find any data in this paper on increase in crop quality for these 25 farmers. There should be some kind of numerical results to evaluate and determine if this program did increase crop quality for those participating farmers. Thus, of relevance to add to this paper is for you to measure and test improvement in “crop quality” or “benefits to farmers” in some manner that quantifies for the reader the importance of this program. The reader needs empirical proof of the success of the program, not just your word.
In general, your discussion is disjoint from results. How did you use your results from the interview, for instance, to adapt the management program? What information did you learn at the biweekly meetings to make adjustments to the program? Examples need to be given in the discussion, thus informing and convincing the readers that using Adaptive Management was beneficial, and beneficial in what ways. Did their program convince other farmers, following the initial 25 farmers, to participate in being certified as wildlife-friendly? Thus, you need to describe in detail how the 25 farms benefited or not from going organic and being certified as wildlife-friendly. I can see where the wildlife benefit, but what about the farmers? More emphasis needs to be placed on this in your revision.
A sentence or two should also be in the Conclusions about how the results of the interviews affected the program’s direction. How did you then adapt? If you were so successful in the adaptive management strategy, there is not enough information to convince the reader of such. This is important information in order to serve as an exemplary case for others to follow. How did the crop quality improved (using a statistical test), by evaluating before and after certification, for example?
Finally, in the pdf, I recommend making a table listing each of the potential pest species and the biological control/s potentially available through a program of this kind to eradicate or reduce the threat of each pest species. This could be in the main paper or an appendix.

Author Response
1. I enjoyed this paper on your project that integrates ecosystem services with agricultural production, demonstrating how wildlife-friendly coffee plantations can effectively protect biodiversity. I have included comments in the attached pdf with corrections and recommendations.
We thank the reviewer for highlighting the novelty and importance of the study, and for providing useful suggestions.
2. My main concern is that the emphasis in the method and result sections on the data analysis from the interviews was barely touched upon in the discussion, and not at all in the conclusions. How does the results of the interview show the benefits of having wildlife on the farm for coffee production? I don't see in this paper these specific results nor do I see where this study actually demonstrated that wildlife benefited from this program.
We understand the concerns of the reviewer. From what we understood from this comment and other comments, the reviewer might have misinterpreted the fact that interviews were only the first stage of the adaptive governance. As specified at lines 129-130, we used the information collected as a first step to identify potential problems for the transition to community-wide use of wildlife-friendly practices. In the paper we then describe the actions we took during different phases of the programme (Table 2) to extend the programme to ~400 coffee farmers. We would thus like to assure that it is evident that the interviews are only the first step we followed, and that our programme involves ~400 farmers and not only the 25 from the interviews. We chose to do interviews with those 25 farmers as they converted to organic farming in 2016 to begin a process to obtain certification based on Indonesian standards, and they did not obtain the organic certification at the time of the interviews. We wanted thus to understand their difficulties, barriers, expectations so that we could have planned to extend organic and wildlife-friendly farming practices to all the coffee farmers in the area. We would like to make clear that the aim of this paper was not to show that our programme benefited wildlife (that would have required a series of wildlife monitoring that would go far beyond the scope of the paper), but rather to describe and assess the steps we followed to develop a wildlife-friendly programme from its conception to reaching certification. We also describe the steps we took to identify problems and find common solutions with key stakeholders. (Lines 84-88). We thus think that highlighting the results of the interviews too much would limit the scope of the paper, as they were preliminary interviews to decide on actions we took after. We would like to point out that the results of the interviews are often considered in all the sections of the discussion. We did not want to repeat the results in the discussion. That said, we have tried to integrate better the results of the interviews with specific links to results, when appropriate.
3. Furthermore, the discussion needs to cover the evolution of the adaptive management process, lessons learned and recommendations to make the adaptive management process successful. You had mentioned that a majority of adaptive management programs had failed, while yours was successful. What did you do differently that made for your success? Did this case study of 25 farmers who went organic and did not hunt vertebrate pests show us that this method improved crop quality? What were the monetary costs to the farmers? How much did they lose to gain a biodiverse ecosystem? I didn’t find any data in this paper on increase in crop quality for these 25 farmers. There should be some kind of numerical results to evaluate and determine if this program did increase crop quality for those participating farmers. Thus, of relevance to add to this paper is for you to measure and test improvement in “crop quality” or “benefits to farmers” in some manner that quantifies for the reader the importance of this program. The reader needs empirical proof of the success of the program, not just your word.
We agree with the reviewer that more details of the criteria used to define the success of our programme can be added, and we have done so in the following way. We initially did not include the data as we wanted to show the phases we followed to develop a wildlife-friendly programme from its conception to reaching certification. We believe that reaching the certification is already a big goal for farmers, and that this, together with the strong participation and other pro-environmental behaviours showed by the farmers (e.g. hunting and littering ban, obtaining shade trees and training from the government), we feel are clear evidence of success. But we have decided to add more results about the pest control as suggested by the reviewer, and we think that these results can improve the paper. We have added further details regarding the other questions of the reviewer in the table and in the discussion.
4. In general, your discussion is disjoint from results. How did you use your results from the interview, for instance, to adapt the management program? What information did you learn at the biweekly meetings to make adjustments to the program? Examples need to be given in the discussion, thus informing and convincing the readers that using Adaptive Management was beneficial, and beneficial in what ways. Did their program convince other farmers, following the initial 25 farmers, to participate in being certified as wildlife-friendly? Thus, you need to describe in detail how the 25 farms benefited or not from going organic and being certified as wildlife-friendly. I can see where the wildlife benefit, but what about the farmers? More emphasis needs to be placed on this in your revision.
We thank the reviewer for this comment and we believe that we have now made clearer throughout the discussion
5. A sentence or two should also be in the Conclusions about how the results of the interviews affected the program’s direction. How did you then adapt? If you were so successful in the adaptive management strategy, there is not enough information to convince the reader of such. This is important information in order to serve as an exemplary case for others to follow. How did the crop quality improved (using a statistical test), by evaluating before and after certification, for example?
We have expanded the conclusion as suggested.
6. Finally, in the pdf, I recommend making a table listing each of the potential pest species and the biological control/s potentially available through a program of this kind to eradicate or reduce the threat of each pest species. This could be in the main paper or an appendix.
We would like to note that we mainly focused on the coffee berry borer as that is the main issue for coffee farmers. Other pests were marginally considered in the programme. We just noted other pests such as civets as they can be perceived negatively by farmers and they may hunt them. We thus think that having a list of pests and possible solutions would be beyond the scope of this paper, but we will certainly consider this aspect for our continuing work on this topic. We considered the comments in the pdf directly in text.
We just specify the response to the following comment as it was difficult just to integrate in text
7. Hunting refers to killing the birds, right? However, I assume the term commercially-sold birds means live birds to be used as pets. I am not sure I am making the connection. Is it that the pet market has reduced the number of birds enough that hunters are not interested in killing birds? Are the birds hunted for food? If your use of hunting is also inclusive of capturing birds for pets then should specify this or instead use word "capturing" in place of "hunting".
Hunting is not just for bushmeat, it is also a hobby, including catching species for personal or commercial pet, coffee or sport trade. We have now made the terminology around catching animals clearer, but also that in Indonesia, the term hunting ban encompasses any kind of catching.
Reviewer 2 Report
This manuscript describes the use of adaptive management to develop a wildlife-friendly programme for sustainabe coffee farming in Java.
The study clearly has merit, given the importance of balancing human and wildlife needs, and will appeal to a broad audience (as an informative case study for those outside of the region, and as a specific prescriptive plan for those within it).
The study was very well presented - the writing, organization, and content were all clear. There are a few very minor grammatical errors, but overall the quality of writing was excellent.
I applaud the authors on their work, both the study itself and its presentation. I am surprised not to have any significant revisions to request, but that speaks to the quality of the work. Well done.
Author Response
This manuscript describes the use of adaptive management to develop a wildlife-friendly programme for sustainable coffee farming in Java.
The study clearly has merit, given the importance of balancing human and wildlife needs, and will appeal to a broad audience (as an informative case study for those outside of the region, and as a specific prescriptive plan for those within it).
The study was very well presented - the writing, organization, and content were all clear. There are a few very minor grammatical errors, but overall the quality of writing was excellent.
I applaud the authors on their work, both the study itself and its presentation. I am surprised not to have any significant revisions to request, but that speaks to the quality of the work. Well done.
We thank the reviewer for appreciating the quality of the paper and for suggesting to accept it. We are also very pleased that the reviewer sees the application of the study more broadly.
Reviewer 3 Report
As scientific papers' reviewer my main objective always is to improving the manuscripts through my comments. But in some cases, the authors they do not give the space. The paper generally is interesting even if -in my opinion- the originality is in low level. From the other hand, authors tried hard the paper have a good presentation -or a good looking- and they did not follow the writing rules of a typical scientific paper (always according my opinion).
In all over manuscript the authors use in the written, first person and without to give clear their role (they were simple researchers? they were members of the adaptive governance? both?)
In my opinion the paper is not clear either well structured. The introduction section is OK but without gives with clarity the objective of the research.
I believe that missing a second section with the theoretical background that will includes between others the legislation and implemented framework (about Indonesia laws and local laws, eco certifications, etc.).
In the study area (subsection 2.1) missing also a map.
I do not observe use of citations who mention to the adaptive management approach.
The structural equation modelling (SEM) was based on data from the 25 local farmers from Panga ban or from the 400 farmers? If the SEM based only on 25 farmers the sample and the results were representative?
Discussion section was used to describe the managing actions and have no any connection with a typical discussion where except results' analyzing there is comparison of them with similar studies. Conclusions are poor also.
In which text' points exist specific proposals in Policies implementation, the originality of research and the study limitations.... and the future directions also.
Author Response
1. As scientific papers' reviewer my main objective always is to improving the manuscripts through my comments. But in some cases, the authors they do not give the space. The paper generally is interesting even if -in my opinion- the originality is in low level. From the other hand, authors tried hard the paper have a good presentation -or a good looking- and they did not follow the writing rules of a typical scientific paper (always according my opinion).
We understand that other papers have followed this approach, but we have not come across many papers that successfully employed the Adaptive Management framework and that showed and discussed the steps followed to promote wildlife-friendly practices and to reach certification. Furthermore, since Wildlife Friendly is a relatively new eco certification, there are few papers that examine the approach around this particular certification. Regarding the writing style, we have reread the paper for grammar and style and again followed the reviewers’ advice. We understand that in some fields, paper structure can vary. From our perspectives, however, as conservation biologists, ecologists and anthropologists, we think the paper has a clear structure, clear methods, clear results, broad discussion and follows the style of other papers in the journal.
2. In all over manuscript the authors use in the written, first person and without to give clear their role (they were simple researchers? they were members of the adaptive governance? both?)
We use “we” when referred to the researchers (authors of the paper). The active voice is always used in our field.
3. In my opinion the paper is not clear either well structured. The introduction section is OK but without gives with clarity the objective of the research.
The other reviewers seemed to understand the aim of the research, and without specific direction it is difficult to make changes that will improve the paper for the reviewer.
4. I believe that missing a second section with the theoretical background that will includes between others the legislation and implemented framework (about Indonesia laws and local laws, eco certifications, etc.).
We have added sentences on the available certifications in Indonesia, indicating the novelty of the WFEN certification.
5. In the study area (subsection 2.1) missing also a map.
We have now added a map as suggested.
6. I do not observe use of citations who mention to the adaptive management approach.
The citations are number 34 and number 40, they are two review papers.
7. The structural equation modelling (SEM) was based on data from the 25 local farmers from Panga ban or from the 400 farmers? If the SEM based only on 25 farmers the sample and the results were representative?
As specified in the methods, the SEM is based on 25 farmers. This sample size included all the farmers that started the process to obtain organic certification in 2016. It is thus representative since it includes 100% of the farmers for whom we wanted to gather information important to start the adaptive governance stage. We reported the model fit to show the stability of the model, and note that our sample size is suitable for modelling of this type.
8. Discussion section was used to describe the managing actions and have no any connection with a typical discussion where except results' analyzing there is comparison of them with similar studies. Conclusions are poor also.
We have now made sure that our results are better discussed in relation to the current literature, and added additional results as asked by another reviewer.
9. In which text' points exist specific proposals in Policies implementation, the originality of research and the study limitations.... and the future directions also.
We are not sure that subheadings with originality of research, research limitations, and future directions are appropriate to our paper as the aim was to describe and assess the steps we followed to develop a wildlife-friendly programme from its conception to reaching certification. We also note that the journal generally does not have these sections. We feel that we have addressed these issues, however within the overall scope of the discussion.
Round 2
Reviewer 1 Report
I appreciate the effort made to revise your paper according to the reviewers’ suggestions. Thanks also for clarifying how the interview process fits into the overall program.
In abstract, you need to add a line about the results regarding the reduction of coffee berry borer between the organic and inorganic farming.
Adding the analysis of the impact of the coffee berry borer on organic versus inorganic farming can make the paper more robust but needs to be better integrated into how these results affect the Adaptive Management Program. To this regard, in your results, you found that organic plantations had significantly higher infestations than inorganic ones and that reduction in infestations was more evident in inorganic plantations. Furthermore, in Figure 4 we see that infestations by this pest species were reduced after treatment in both types of plantations, with better results for the inorganic plantations. You need to expound on this in the discussion, particularly addressing the contradiction that organic plantations are the way to go in terms of wildlife and ecosystem biodiversity, yet we do not see as great a reduction in the main pest species. Won’t farmer investment in organic farming be affected by your results that show that management of coffee berry borers are less effective on organic farms than inorganic ones? You must address this discrepancy between the benefits of organic farming with the less effective control of this pest species. It is important that there was a reduction in pests on organic farms but at the same time the significance in the reduction between the two types of farming needs to be addressed. What actions do you recommend to be implemented to encourage farmer investment despite less effectiveness of pest control methods in organic coffee farming?
Author Response
I appreciate the effort made to revise your paper according to the reviewers’ suggestions. Thanks also for clarifying how the interview process fits into the overall program.
We thank the reviewer for appreciating our effort. The suggestions provided really improved the paper.
In abstract, you need to add a line about the results regarding the reduction of coffee berry borer between the organic and inorganic farming.
The information is now added.
Adding the analysis of the impact of the coffee berry borer on organic versus inorganic farming can make the paper more robust but needs to be better integrated into how these results affect the Adaptive Management Program. To this regard, in your results, you found that organic plantations had significantly higher infestations than inorganic ones and that reduction in infestations was more evident in inorganic plantations. Furthermore, in Figure 4 we see that infestations by this pest species were reduced after treatment in both types of plantations, with better results for the inorganic plantations. You need to expound on this in the discussion, particularly addressing the contradiction that organic plantations are the way to go in terms of wildlife and ecosystem biodiversity, yet we do not see as great a reduction in the main pest species. Won’t farmer investment in organic farming be affected by your results that show that management of coffee berry borers are less effective on organic farms than inorganic ones? You must address this discrepancy between the benefits of organic farming with the less effective control of this pest species. It is important that there was a reduction in pests on organic farms but at the same time the significance in the reduction between the two types of farming needs to be addressed. What actions do you recommend to be implemented to encourage farmer investment despite less effectiveness of pest control methods in organic coffee farming?
We thank the reviewer for pointing out this issue as we did not notice that the information provided was confusing. The plantations classified as inorganic are the ones that started as inorganic and turned into organic when joining the programme. They are a subset of plantations from farmers that joined the programme and are helping to co-produce actions. They had even better results than the farmers that started as organic, probably because using organic for them was new and they were really keen to learn new practices, especially since we brought experts from different universities and the government to do the training. Having tangible results (e.g. evident reduction in coffee berry borer and thus in coffee quality and productivity) was key to persuade them to continue our programme. We clarified these aspects in the abstract, results, discussion, conclusion.
Reviewer 3 Report
Some of my comments were taken into account by the authors. However, in my opinion the manuscript has elements of a technical report that describe the course of a project than the form and the content of a research paper.
Author Response
We thank the reviewer for acknowledging that we partially satisfied his concerns. We strongly believe that our manuscript is not a technical report, and we feel that Land journal published many papers that are similar in form and content to our paper. We thus cannot proceed with further edits in the form and content unless further comments are provided.